# Study on Preparation and Properties of Super Absorbent Gels of Homogenous Cotton Straw-Acrylic Acid-Acrylamide by Graft Copolymerization

**DOI:** 10.3390/gels11080583

**Published:** 2025-07-28

**Authors:** Jun Guo, Jing Shi, Lisheng Xu, Xingtao Zhang, Fangkai Han, Minwei Xu

**Affiliations:** 1College of Biology and Food Engineering, Suzhou University, Suzhou 234000, China; guojun@ahszu.edu.cn (J.G.);; 2Department of Plant Sciences, North Dakota State University, Fargo, ND 58108, USA; minwei.xu@ndsu.edu

**Keywords:** homogenous cotton straw, acrylic acid, water absorption ratio, acrylamide, graft polymer

## Abstract

To rationally utilize and develop agricultural waste products, this research involved the synthesis of degradable high water-absorbing resin through the graft copolymerization of cotton straw (CS) with monomers. Among them, acrylic acid (AA) and acrylamide (Am) are used as grafting copolymer monomers, cellulose in the straw serves as the network framework, and MBA acts as the crosslinking agent. ^60^Co gamma rays as initiators. Different concentrations of alkaline solution were used to dissolve the cellulose in the straw. Single-factor and orthogonal experiments were conducted to optimize the experimental conditions. various analytical methods such as thermogravimetric analysis (TG), X-ray crystallography (XRD), infrared spectroscopy (IR), and scanning electron microscopy (SEM) were employed to characterize the structure and properties of the product. ^60^Co gamma rays as initiators, can reduce the pollution caused by chemical initiators and lower energy consumption. Through this research, agricultural waste can be effectively utilized, reducing environmental pollution, lowering industrial energy consumption, and synthesizing degradable and environmentally friendly high-absorbent resins. The product can be applied to agricultural water retention agent, fertilizer controlled release agent and other aspects.

## 1. Introduction

Super absorbent gels (SAG) is a new type of functional polymer material of hydrogel containing a large amount of water with a three-dimensional network structure [1]. It can absorb and retain water thousands of times its own weight [2]. It is widely used in various fields such as hygiene products, daily life, agricultural and forestry protection, household items, food preservation, construction industry, and medical materials [3]. The traditional methods for preparing SAG are generally chemical synthesis [4]. In recent years [5,6], there have been reports on preparing SAG by graft copolymerization using natural polymers such as starch and cellulose. These super absorbent gels which based starch and cellulose have good biodegradability, with the gradual strengthening of people’s environmental awareness, the development and research of such super absorbent resins are showing a rapid development trend [7]. These super absorbent gels which based starch and cellulose have different advantages and disadvantages compared to chemical synthetic gels. As a non-ionic absorbent material, super absorbent gels which based starch and cellulose has better salt tolerance and faster absorption speed than ionic absorbent gels [8,9,10]. Therefore, this study intends to select CS which rich in natural cellulose as the graft copolymerization object to prepare super absorbent gels.

Starch and cellulose are natural macromolecular materials [11,12,13], starch and cellulose were semi-crystalline biomacromolecule containing multiple hydroxyl hydrophilic groups, the chemical bond of starch is α-glycosidic bond, the chemical bond of cellulose is β-glycosidic bond, are natural polysaccharide polymer that possess good biodegradability and biocompatibility, and environmentally friendly [3,6,14]. SAG of natural polysaccharide polymer decompose naturally, reducing the environmental impact associated with synthetic polymers, which can persist in the environment for a long time. Both starch and cellulose are derived from renewable resources (such as corn, potatoes, and plant fibers), making them more sustainable compared to petroleum-based synthetic SAG. Starch and cellulose are often less expensive to produce compared to synthetic alternatives, particularly when sourced from agricultural byproducts [15]. They are non-toxic and generally recognized as safe, making them suitable for applications in food, agriculture, and personal care products, compatible with biological systems, making them suitable for medical and agricultural applications where interaction with living organisms is a concern. SAG of natural polysaccharide polymer can be modified chemically or physically to enhance their properties, such as improving water absorption capacity or creating composite materials with other substances [16]. The production processes for gels of natural polysaccharide polymer can require less energy compared to the synthesis of polyacrylic acid, contributing to a lower carbon footprint [10]. Starch and cellulose can offer additional functional properties, such as thickening, gelling, or film-forming abilities, which can be beneficial in various applications [17]. The synthesis of superabsorbents can sometimes produce harmful by-products, while natural polymers typically do not have this issue [7,11]. Compared with gels of polyacrylic acid, gels of starch and cellulose has a series of advantages, such as low production cost, simple process, high production efficiency, strong water absorption ability and long product shelf life, etc., which has become the current research hotspot in this field [5,6,16,18].

The existing studies have focused more on the production of SAP using starch, while there are relatively fewer studies on the production of such resins using cellulose from straw fibers [5,18,19]. The possible reason is that the cellulose in straw contains a relatively high content of β-glycosidic bonds and hydrogen bonds, which form large number of crystalline regions, making cellulose difficult to dissolve in water. Make the water absorption performance of the SAG of cellulose-based relatively lower than that of the starch-based high water-absorbing gels. Moreover, the chemical small molecules such as acrylic acid and acrylamide are also unable to enter the interior of cellulose to undergo graft copolymerization reactions [20].

Cotton straw are a kind of by-product with huge yield from cotton cultivation and an important source of biomass materials [21]. These stalk materials are natural lignocellulosic resources, rich in lignin, hemicellulose and cellulose, as well as potassium, phosphorus, nitrogen and other elements [22]. If they can be fully recycled and reused, they will become valuable resources [23]. Due to the lack of perfect recycling channels, insufficient processing technologies and the need for exploration in application, at present, the main treatment methods for cotton stalks are incineration or burial for land reclamation [24]. Incinerating cotton stalks not only poses great safety hazards but also generates about 250 million tons of carbon dioxide and 28 million tons of nitrogen oxides every year [25,26]. The dust emitted into the air contains more than 100,000 tons of alkylated polycyclic aromatic hydrocarbons (APAHs) and polycyclic aromatic hydrocarbons (PAHs) particles, which will aggravate the haze pollution [27]. Studies have reported that soil amendment with cotton stalks cannot quickly cultivate soil due to the problem of decomposition degree and may cause pollution to the entire layer of soil [28]. Many researchers have attracted attention to seeking other treatment and application methods for cotton stalks [29].

Cellulose is the most widely distributed and abundant natural macromolecule in nature. It has a large degree of polymerization, a stable structure, and relatively stable mechanical properties. It is a renewable resource, inexpensive, biocompatible, and has good degradability. It is one of the research directions of green chemistry and has great development and utilization value. Cellulose is composed of multi-dispersed linear glucose polymer chains. At the same time, due to a large number of hydrogen bonds, it forms a supramolecular structure with high crystallinity, which makes it insoluble in most reagents and limits its application. For example, when using wheat straw to produce water-absorbing gels, the cellulose in the wheat straw is not effectively de-crystallized, and the steric hindrance during grafting increases, thereby limiting the performance of the resin [13]. Therefore, establishing an effective, green, and environmentally friendly cellulose dissolution system will improve the utilization efficiency of cellulose. However, there are few reports on the graft copolymerization of gels reaction of straw cellulose in the dissolved state. This study will seek suitable cellulose dissolution reagents to dissolve cellulose in cotton straw.

Graft copolymerization can enhance properties such as mechanical strength, thermal stability, chemical resistance, and water absorption, making the resulting materials more suitable for diverse applications [8,20]. This method can be conducted without the use of harmful solvents or chemicals, making it a cleaner and more environmentally friendly approach compared to traditional chemical grafting methods. Graft copolymerization induced by radiation of ^60^Co is a technique used to modify polymers by attaching side chains (graft chains) onto a main polymer backbone through the using of radiation of gamma rays [30,31]. The radiation generates highly reactive radicals, which can efficiently initiate the grafting process, leading to a more effective and rapid reaction compared to some traditional methods [32]. Radiation grafting processes can be scaled up for industrial applications, making it practical for large-scale production of modified polymers [33]. By using different monomers for grafting, materials can be tailored to have multiple functional properties, such as antimicrobial activity, hydrophilicity, or specific adsorption capabilities. Radiation-induced graft copolymerization allows for the incorporation of a wide variety of side chains, enabling the modification of polymer properties to meet specific application requirements. The degree of grafting can be controlled by adjusting the radiation dose and the reaction conditions, allowing for the fine-tuning of the material properties. Since graft copolymerization can occur in solid or aqueous phases, it can minimize the presence of unreacted monomers in the final product, leading to safer materials for various applications [34]. Grafted polymers can improve compatibility between different phases or components in composite materials, enhancing overall performance and functionality, produced through radiation can be used in various fields, including biomedical applications (e.g., drug delivery, tissue engineering), agriculture (e.g., superabsorbent materials), and packaging [35]. In summary, radiation-induced graft copolymerization offers a versatile, environmentally friendly, and effective method for modifying polymers, resulting in materials with enhanced properties for a wide range of applications [36].

In this study, cotton stalks were dissolved in different concentrations of alkaline solution to obtain homogeneous stalk solutions without crystalline regions. This process enabled the large molecular fibers in the stalks to undergo graft copolymerization with monomer molecules. The copolymerization reaction was initiated by cobalt radiation, which saved chemical initiators and reduced pollution. The graft copolymerization of soluble cellulose of CS with acrylic acid (AA) and acrylamide (Am) were used to produce a SAG, N,N′-Methylene-bisacrylamide (MBA) was used as crosslinking agent, the SAG properties and structure were measured, hoping to obtain the SAG with superior performance.

## 2. Results and Disscusion

### 2.1. Effect of SAP on Water Absorption of CS with Different Pretreatment

Such as Table 1, formulate four kinds of NaOH/urea systems with different proportional concentrations: NaOH (6 wt%)/urea (4 wt%), NaOH (5 wt%)/urea (5 wt%), NaOH (4 wt%)/urea (6 wt%), NaOH (3 wt%)/urea (7 wt%), the CS pow-der was divided into 4 equal parts and put into 4 groups of solution, stirring for 15 min and then put into re-frigerator (5 °C), stirring for 12 h. After removal, the clear solution CS1, CS2, CS3, CS4 of cellulose can be obtained, It can be seen from Table 1 that in the NaOH/urea system, when the concentration ratio of sodium hydroxide to urea was NaOH (5 wt%)/urea (5 wt%), the grafted hyper absorbent resin obtained from prepare CS2 had higher grafting efficiency and better absorption effect of various solutions. The possible reason is that the ratio solution has a good effect on the decrystallization of CS, and does not cause the degradation of cellulose, so that the resin with higher crosslinking degree can be obtained. CS powder was processed by high pressure alkaline cooking (150 °C, 0.6 MPa, 30 min), then degraded by nitric acid with concentration of c(HNO_3_) = 1 mol/L at 100 °C for 30 min, and then filtered and washed to obtain cellulose CS5, CS powder was soaked in 10% NaOH aqueous solution for 24 h, m (straw):m (aqueous solution) = 1:12, washed and dried to obtain cellulose CS6 for use, 7 g CS powder, then 70 mL dilute sulfuric acid with a mass fraction of 0.75%, controlled at about 100 °C, heated in a water bath for 2 h, then filtered and dried to obtain cellulose CS7.

It can be well seen from Table 2 that the cellulose graft copolymer resin in NaOH (5 wt%)/urea (5 wt%) aqueous solution had higher water absorption multiple than the SAG made from CS treated by other methods, showing higher salt absorption water and higher grafting efficiency. This indicated that cellulose treated with NaOH/urea could be grafted and copolymerized better to form a SAG with good water absorption. The possible reason was that the neat and dense hydrogen bond force of cellulose was changed by NaOH/urea, and the molecules changed from orderly and tight arrangement to irregular and loose structure, and the monomer molecules entered the cellulose for graft copolymerization. A uniform three-dimensional network structure with high cross-linking degree was formed, and there were more adsorption sites inside the resin. Because the NaOH (5 wt%)/urea (5 wt%) system itself contained urea, SAG2 of itself contained urea, so the absorption of artificial urine was slightly lower than that of other resins made of straw powder. Among SAG5, SAG6 and SAG7, the water absorption ratio of SAG6 is higher. It is possible that the molecular structure of wheat straw powder also changes under high temperature and high pressure alkaline distillation and concentrated acid. However, such treatment also breaks the glucoside bond on the cellulose molecular chain, resulting in degradation of cellulose macromolecules, and thus low cross-linking degree.

The alkaline solution contains highly polar hydroxide ions and sodium ions. It can break the hydrogen bonds between cellulose molecules, causing cellulose to lose its crystalline structure, dislodge the supermolecular coiling structure of cellulose, and separate the long chains of molecules. This results in the swelling and dissolution of cellulose which fiber transfers to dissolved polymer, improve the efficiency of graft copolymerization [37].

### 2.2. Single Factor Experiment

#### 2.2.1. Influence of Ratio of Monomer to CS2 on Water Absorption of SAP

Based on preliminary pre-experiments and references [1,16], as the dose is 4.5 kGy, the addition of MBA is 1.2 % (*m*/*v*), the AA neutralization is 80%, the rate of AA to Am is 1.5 [1]. From Figure 1a, by changing the mass ratio of monomer to CS2 and keeping other factors unchanged, the influence of these factors on the water absorption of butt copolymer was studied. When m (AA + Am):m (CS2) = 7:1, the water absorption ratio of the graft copolymer reaches the maximum. When the mass ratio of monomer to cellulose increases successively, the water absorption ratio of the hydrogel increases first and then decreases, and the water absorption ratio decreases more sharply with the rapid increase of the mass ratio. This may be because the amount of CS2 directly affects the crosslinking density of SAG. When the amount of CS2 is too low, it is not easy to form cross-link, the cross-linking degree is low, the reaction rate is slow, and the water absorption rate is low. When the amount of CS2 is too high, the molecular weight of CS2 is not uniform, and it is easy to react, resulting in excessive cross-linking, which also affects the water absorption rate of hydrogel.

#### 2.2.2. Influence of Mass Ratio of AA and Am on Water Absorption Ratio of SAP

Based on preliminary pre-experiments and references [1,16], as the rate of monomer to CS2 weight ratio is 7, the dose is 4.5 kGy, the addition of MBA is 1.2% (*m*/*v*), AA neutralization is 80% [1]. From Figure 1b, by changing the mass ratio of monomer to CS2 and keeping other factors unchanged, when the ratio of Am to AA is 1.5, the distill water absorption rate (Q_d_) and salt water absorption rate (Q_s_) of the hydrogel is the highest, while the salt water absorption rate keeps increasing. The possible reason is that with the increase of the amount of Am, the amide group of the graft copolymer increases correspondingly. On the one hand, because the amide group is a non-ionic hydrophilic group, it is bonded with water by hydrogen bond. The change of ionic strength in electrolyte has little effect on its water absorption capacity, and it is more absorbent to electrolyte containing solution or other non-pure water solution, so CS2-Am-AA graft gels has strong salt tolerance. On the other hand, the possible reason is that with the increase of amide base sites, the group decreases, and the amide group does not participate in the formation of osmotic pressure, but only the group participates in the formation of osmotic pressure. Thus, the osmotic pressure inside and outside the colloid is reduced, which causes the absorbent resin to absorb deionized water

#### 2.2.3. Influence of Acrylic Neutralization on Water Absorption of SAG

Based on preliminary pre-experiments and references [1,16], as the dose is 4.5 kGy, the addition of MBA is 1.2% (*m*/*v*), the rate of m (monomer):m (CS2) = 7:1, the rate of AA to Am is 1.5 [1]. As shown in Figure 1c, when the neutralization degree of (AA) is within the range of the range of 50% to 80%, before reaching the optimal value, the reaction speed of polymerization is difficult to control, so it is easy to produce a large number of highly cross-linked polymer homopolymers, resulting in the water absorption capacity of the resin is affected, and the water absorption performance is deteriorated. With the increasing of neutralization degree, the graft copolymerization reaction rate of CS2 and AA was accelerated, and the carboxylic sodium group was also increased, so the water absorption capacity of hydrogel was improved. When the neutralization degree of AA exceeds 80%, resulting in the generation of free radicals is affected, because of hydrogel reaction will also be affected, resulting in a decrease in the water absorption capacity of the hydrogel, it can be seen from the figure that the neutralization degree of acrylic monomer is about 80% of the gels has the best water absorption performance. When the neutralization degree of AA exceeds 90%, resulting in a decrease for the sodium chloride solution absorption capacity of the hydrogel, it may be speculated that lower pH hydrogel are more conducive to the absorption of sodium chloride ions.

#### 2.2.4. Influence of Mass Ratio of Crosslinking Agent to CS2 on Water Absorption Ratio of SAG

Based on preliminary pre-experiments and references [1,16], as the rate of monomer to CS2 weight ratio is 7, the dose is 4.5 kGy, AA neutralization is 80%, the rate of AA to Am is 1.5. From Figure 1d, the addition of MBA is 1.2% (*m*/*v*), the distill water absorption rate (Qd) and salt water absorption rate (Qs) of the product is the highest, and as the mass fraction of MBA gradually increases, the water absorption ratio of SAG first increases and then decreases, and as When the addition of MBA exceeds 1.2%, the Qd and Qs of SAG decreases. This may be due to the fact that MBA and Am, will directly affect the spatial network structure and crosslinking degree of the SAG and thus affect the water absorption. When the amount or mass fraction of N=N, the spatial network structure has a low crosslinking degree, which shows a semi-water-soluble state and a low water absorption rate. When the amount or mass fraction of the hydrogel is too large, it will lead to too many cross-linking points in the gels space network, and the cross-linking points in the formed space network structure are too dense, which will reduce the amount of water contained and the water absorption ratio will be reduced accordingly.

### 2.3. Orthogonal Experiment

Through a single factor experiment, choose three factors, the ratio of monomer to CS2, the neutralization degree of AA, the ratio of AA to Am. The orthogonal experiment of three factors and three levels was designed, as shown in Table 3, through the three-factor and three-level orthogonal experiment, it is found that the key factor affecting the water absorption rate is the ratio of AA to Am, followed by the ratio of monomer to CS2, and finally the ratio of acrylamide to acrylic acid. The ratio of acrylamide to acrylic acid is the most important factor affecting the rate of brine absorption of hydrogel. The initial stage of the reaction is chain initiation, chain growth reaction, and crosslinking occurs in the later stage of the reaction. When the ratio of AA to Am is too low, there is less amide group, the water absorption power of hydrogel depends on the osmotic pressure formed, and the hydration mode mainly depends on ionic bond. With the increase of the ratio of acrylamide, the combination of water and mode increases in the form of hydrogen bond, and the water absorption in brine increases. Therefore, the ratio of AA to Am is more appropriate at 1.5. For the A2B3C3 configuration, the absorbency of hydrogel reaches 95 g/g in a 0.9-wt% NaCl solution; and A2B2C3 for the distilled water absorbency of 984 g/g. The effects of distilled water absorbency can be arranged as B > A > C (Table 1), while that for 0.9-wt% NaCl solution it was C > A > C (Table 2). Consequently, the optimal combination was determined to be A2B2C3 for distilled water and A2B3C3 for 0.9-wt% NaCl solution. Under these parameters, CS2-g-AA-co-Am was synthesized without the need for heating or nitrogen protection, yielding an absorbency of 824 g/g for distilled water and 95 g/g for the NaCl solution.

### 2.4. Structure and Morphology Analysis of SAG

#### 2.4.1. IR Spectra of Sample

The FT-IR spectra of CS and SAG2 are shown in Figure 2, it can be seen that both rice straw and grafted copolymers have an OH stretching vibration peak at 3446 cm^−1^ on the cellulose sugar ring, and an antisymmetric stretching vibration peak of -CH_2_ at 2924 cm^−1^. From the Figure 2a, it is the FT-IR spectrum of CS, the 1752 cm^−1^ position is the stretching vibration peak of the epoxy carbonyl group in cellulose, the 1602 and 1641 cm^−1^ positions are the characteristic absorption peaks of the carbonyl group in pentenyl acetal, and the 1418 and 1375 cm^−1^ positions are the stretching vibration peaks of the β-1,4-glycosidic bond on the cellulose chain. Therefore, the main component of rice straw is cellulose. In the FT-IR spectrum of the SAG, the 1688 cm^−1^ position is the stretching vibration peak of the carbonyl group of the SAG2 acrylic group, the 1465 and 1426 cm^−1^ positions are the characteristic absorption peaks of the amide group of acrylamide, and the peak at 1285 cm^−1^ indicates that the β-1,4-glycosidic-O-atom rocking vibration peak of cellulose in the SAP becomes weaker, indicating that the content of cellulose in the SAG2 is lower than that in the CS. Thus, it is proved that when CS cellulose and monomer active small molecules undergo co-irradiation, graft copolymerization occurs, and the product is the SAG2 of CS fiber-acrylic acid-acrylamide [2,38].

#### 2.4.2. XRD Analysis

Cellulose is a natural high molecular weight polysaccharide compound. The weak intermolecular hydrogen bonding interactions and polar charge distribution lead to the formation of a supramolecular form [39]. After being treated with the corresponding chemical or physical methods, the supramolecular entanglement state within the molecules changes, the hydration capacity is enhanced, and after graft copolymerization with small chemical molecules, the supramolecular structure is further stretched, making the amorphous regions richer [40]. Therefore, the XRD method is used to macroscopically describe this phenomenon. By the Figure 3a, X-ray diffraction (XRD) patterns of CS, CS2 and SAG are shown. The CS showed a narrow peak around 2θ = 23°, which indicated that CS contains a relatively large number of hydrogen bonds, CS2 contains fewer hydrogen bonds. While SAG formed a wider diffusion peak here, indicating that the graft copolymerization broke the hydrogen bonds, and the graft copolymer was amorphous, forming a resin [41]. 

#### 2.4.3. Thermogravimetric Analysis of Sample

As the Figure 3b shows, thermogravimetric analysis was performed on CS and CS2 after drying at 5.00~7.00 mg. It can be seen from the figure that the loss of free water and bound water in the sample led to a relatively mild thermogrgravity phenomenon for both CS and CS2 below 200 °C. The curve of CS was not smooth enough, indicating that the molecular weight of CS was inconsistent. The initial thermal decomposition temperature of CS and CS2 was about 230 °C, and the decomposition was complete at 450 °C. The CS2 decomposed completely at 550 °C. Therefore, the first decomposition of CS under the same conditions indicates that the thermal stability of CS2 is better than that of CS, which may be due to the decrease of intermolecular hydrogen bond energy and the increase of thermal stability of grafted copolymer.

#### 2.4.4. SEM Analysis of SAG

SEM was used to analyze the surface morphology of the SAG2, from Figure 4, it can be seen that the graft copolymer is a loose three-dimensional network, no regular crosslinking structure, its surface area is extremely large, which determines the water absorption rate of the hydrogel is very fast. It is this structure that water molecules quickly penetrate into the interior and are stored through the hydrogen bonding of hydrophilic groups, thus giving such materials high water absorption properties and high water absorption speed.

## 3. Conclusions

Through the single factor method and the structural property analysis of the corresponding products, it can be known that the process parameter conditions as ratio of monomer to CS2 is recommended 7:1; The mass percentage of crosslinker is 1.2%; The water absorption rate of hydrogel is the best when the neutralization degree of acrylic acid is 80%. If used in medical or agricultural treatment of higher ion concentration solutions, it is recommended to use a 90% neutralization system.The ratio of monomer to cellulose is recommended 7:1, the ratio of Am to AA is suggest 1.5, the absorbency of 824 g/g for distilled water and 95 g/g for the NaCl solution for SAG.

Straw cellulose is a natural macromolecular polysaccharide, which is composed of glycosidic bonds linking sugar units into chain-like polymers. As a resin matrix, it is easily degradable because the formation of a three-dimensional network structure requires the cross-linking of small molecule monomers. With less cellulose usage, it will be used as a fertilizer controlled-release agent and water retention agent for field experiments in the later stage. The rate of salt water absorption in this experiment reflects to a certain extent that it can maintain a certain amount of ionic water in the field environment.

## 4. Materials and Methods

AA (analytical grade, without further purification), Am (analytical grade) and MBA (chemically pure) were from Shanghai Chemical Reagent Corporation (Shanghai, China). CS was obtained from suburban of Anhui Chizhou (Chizhou, China). All other commercially available solvents and reagents were analytical grade, and were used as received. The salt water was prepared by dissolving 0.90 g sodium chloride (NaCl) in 99.10 mL distilled water. The artificial urine was prepared by dissolving 1.94 g urea, 0.08 g NaCl, 0.06 g calcium chloride (CaCl_2_) in 97.81 mL distilled water. Unless otherwise specified, double distilled water (with electrical conductivity of 6.40 μS/cm) was used throughout the experiment. ^60^Co-gamma irradiation source, Hefei (National) Forestry Irradiation Center, activity 200,000 curies, single-grid plate-shaped, absorption measurement rate using dichromate dose calibration (Hefei, China).

### 4.1. Pretreatment of CS

Firstly, formulate four kinds of NaOH/urea systems with different proportional concentrations: NaOH (6 wt%)/urea (4 wt%), NaOH (5 wt%)/urea (5 wt%), NaOH (4 wt%)/urea (6 wt%), NaOH (3 wt%)/urea (7 wt%), the CS powder was divided into 4 equal parts and put into 4 groups of solution, stirring for 15 min and then put into refrigerator (5 °C), stirring for 12 h. After removal, the clear solution CS1, CS2, CS3, CS4 of cellulose can be obtained by rising the temperature to room temperature, and the ethanol precipitate of the cellulose can be taken for testing [42].

According to Figure 1, CS powder was processed by high pressure alkaline cooking (NaOH, 6 wt%, 150 °C, 0.6 MPa, 30 min), then degraded by nitric acid with concentration of c(HNO_3_) = 1 mol/L at 100 °C for 30 min, and then filtered and washed to obtain cellulose CS5 [43].

CS powder was soaked in 10% NaOH aqueous solution for 24 h, m (straw):m (lye) = 1:12, washed and dried to obtain cellulose CS6 for use [37].

7 g CS powder, then 70 mL dilute sulfuric acid with a mass fraction of 0.75%, controlled at about 100 °C, heated in a water bath for 2 h, then filtered and dried to obtain cellulose CS7 [44].

### 4.2. Preparation of Graft Copolymer

Weigh a certain amount of CS1, CS2, CS3, CS4, CS5, CS6 and CS7 powder and place it in a beaker, stir and add a certain amount of AA and AM (cellulose:Monomer 1:7, AA:AM is 3:2, AA neutralization 80%) and crosslinking agent MBA (N,N′-methylenebisacrylamide, 1.2%, *m*/*v*) were fully stirred and sent to the irradiation workshop for irradiation polymerization. According to Guo Jun et al. [1], the irradiation dose was set at 4.5 kGy, the irradiation dose was set at 1.5 kGy/h, and the products were SAG1, SAG2, SAG3, SAG4, SAG5, SAG6, SAG7 [45].

### 4.3. Preparation of Composites

According to Figure 2, the SAG preparation: Briefly, In the CS solution, a certain amount of AA (with a certain neutralization degree of 80%) and AM were added while stirring. Stir for 10 min, then add the polymerization inhibitor (FeSO_4_·7H_2_O, 0.2%, *m*/*v*), the cross-linking agent (MBA), and stir evenly. After that, it is sent to the irradiation workshop for irradiation polymerization. This results in the production of the crude product. The crude product is dried at 60 °C, ground, and extracted with ethyl ether as the solvent using a Soxhlet extractor for 4 h. The monomers that have not reacted with the homopolymer are removed. After the ethyl ether evaporates, it is dried at 40 °C until a constant weight is achieved, thus obtaining SAG [9].

### 4.4. Water Absorption of SAP

The water absorption rate of of the SAG was measured at ambient temperature (25 °C). Sample was dipped in the distilled water or saline solution to reach the adsorption equilibrium. Excess water is removed through a 120-mesh sieve. After SAG absorbing water and achieving a stable weight, the value is read. The salt water absorption rate (Q_s_) and distill water absorption rate (Q_d_) of the SAG are calculated using the following Formula (1), each sample was tested three times [5,19], standard error terms for Q from covariance (ANCOVA) using SPSS v. 19.0 (IBM Inc., Armonk, NY, USA). (1)$$Q=\frac{m_{2}-m_{1}}{m_{1}}$$
where *m*_1_ is weight of the dry sample, *m*_2_ is weight of the swelling sample (g). The Q was expressed as gram of water per gram of sample (g/g).

### 4.5. Structure and Morphology Analysis

The FTIR spectra were collected using the instrument (Thermo Nicolet, NEXUS, Madison, WI, USA), and potassium bromide was used for the pressing of the sampleMorphological structure analysis was conducted using a scanning electron microscope (JSM-5600LV SEM from JEOL, Ltd., Tokyo, Japan). Take the SAG powder (passing through a 200-mesh sieve), magnify it 500 times, and perform scanning electron microscopy in a dry state. The sample was coated with a thin layer of gold film at an acceleration voltage of 15 kV. The comparison experiment on the thermal stability of resin and straw was conducted using a thermal anal-ysis instrument (Netzsch STA-449C thermogravimetric analyzer, TGA, Selb, Bavaria, Germany), temperature range: 25–550 °C, the rate of heating of 10 °C/min. Dry nitrogen flow rate of 40 mL/min. Powder X-ray diffraction (XRD) measurements were carried out using Haoyuan DX 2700 (Beijing, China) with Cu K radiation (λ = 1.5406 Å). The diffractometer was operated at 40 kV and 40 mA. The data were collected in the 5 to 50° (2 min) at a step size of 0.02°.

## Data Availability

The data generated during the present study are available from the corresponding author upon reasonable request.

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
