# Peer review of "Study on Preparation and Properties of Super Absorbent Gels of Homogenous Cotton Straw-Acrylic Acid-Acrylamide by Graft Copolymerization"

_gels, 2025, doi:10.3390/gels11080583_

Round 1

Reviewer 1 Report

Comments and Suggestions for Authors
  1. The gels produced are proposed for agricultural and industrial applications. Have field or practical tests (such as soil water retention or controlled fertilizer release tests) been conducted to evaluate the actual performance of these gels?
  2. Biodegradability of the produced gels is cited as a key advantage. Have experimental tests (such as biodegradation tests in soil or compost conditions) were conducted to assess the rate and extent of biodegradation?
  3. It has been claimed that gamma irradiation is less polluting and energy-intensive than traditional chemical methods. Is there quantitative data (such as comparing carbon emissions or energy consumption) to support this claim?
  4. What are the details of the thermogravimetric (TG), X-ray diffraction (XRD), and scanning electron microscopy (SEM) results? Why are these data not fully presented in the text of the article? Can you provide relevant graphs or images to support the results?
  5. How exactly was the process of homogenizing cotton straw (CS) with alkali/urea solution (5 wt% NaOH and 5 wt% urea) carried out?

Author Response

Point 1. The gels produced are proposed for agricultural and industrial applications. Have field or practical tests (such as soil water retention or controlled fertilizer release tests) been conducted to evaluate the actual performance of these gels?

Answer: Thank you for your suggestion, our research team is currently exploring methods for encapsulating fertilizers and releasing them in the fields. This current paper focuses on the synthesis, performance and structural studies of the fertilizers.

Point 2. Biodegradability of the produced gels is cited as a key advantage. Have experimental tests (such as biodegradation tests in soil or compost conditions) were conducted to assess the rate and extent of biodegradation?

Answer: Thank you for your suggestion, our research team intends to conduct the next phase of research on how this highly absorbent resin naturally degrades in soil. Currently, this paper focuses on synthesis, performance and structural studies.

Point 3. It has been claimed that gamma irradiation is less polluting and energy-intensive than traditional chemical methods. Is there quantitative data (such as comparing carbon emissions or energy consumption) to support this claim?

Answer:

Irradiation catalysis for graft copolymerization saves chemical initiators and eliminates the residue of chemical initiators. Relevant literature can be cited in the revised manuscript:

“Benamer, S.; Mahlous, M.; Tahtat, D.; Nacer-Khodja, A.; Arabi, M.; Lounici, H.; Mameri, N., Radiation synthesis of chitosan beads grafted with acrylic acid for metal ions sorption. Radiation Physics and Chemistry 2011, 80, (12), 1391-1397.

  1. Biswal, J.; Kumar, V.; Bhardwaj, Y.; Goel, N.; Dubey, K.; Chaudhari, C.; Sabharwal, S., Radiation-induced grafting of acrylamide onto guar gum in aqueous medium: Synthesis and characterization of grafted polymer guar-g-acrylamide. Radiation Physics and Chemistry 2007, 76, (10), 1624-1630.”

Reviewer 2 Report

Comments and Suggestions for Authors

The manuscript describes a study on the preparation and characterization of superabsorbent polymers (SAPs) synthesized from cotton straw, acrylic acid, and acrylamide using 60Co gamma irradiation. The research focuses on sustainable SAP production by utilizing agricultural waste and evaluates potential applications in agriculture and medicine. The experimental approach includes both single-factor and orthogonal experiments to optimize SAP properties, and various analytical techniques (FTIR, XRD, TG, SEM) are used to support the findings. The study highlights the use of agricultural waste, a solvent-free radiation grafting method, and thorough material characterization. Results indicate high water absorbency and potential for practical application.

The manuscript would benefit from more detail on experimental reproducibility, statistical analysis, and a clearer comparison with other SAPs, especially in terms of biodegradability, cost, and environmental impact throughout synthesis and product use:

  1. Ensure that the quality of Figure 2 is clearly labeled i.e FTIR identifying the key functional groups, a more detailed discussion of the peak shifts and changes in intensity upon superabsorbent polymers (SAPs) formation would be beneficial. Figure 3. (a) XRD indexing and showing crystalline phases and Include the International Center for Diffraction Data (JCPDS) in the crystalline phases. Refer to(doi.org/10.1002/jctb.6657)and is recommended for citation.
  2. Figure3.(b) TG and DTG could be improved by better labeling.
  3. Could you provide detailed information on the reproducibility of SAP synthesis process, including the number of independent replicates performed and any statistical analyses undertaken?
  4. How does the long-term biodegradability and environmental safety profile of your SAPs compare to conventional synthetic alternatives, particularly regarding residual monomers and irradiation processes?
  5. Can you elaborate on the scalability and economic feasibility of utilizing cotton straw as a raw material, considering supply chain logistics and industrial-scale production challenges?
  6. How do the mechanical properties (such as gel strength and elasticity) of your SAPs compare to those of commercially available products, especially under varying environmental conditions?
  7. Have you assessed the SAPs' practical performance, stability in real-world agricultural or medical settings beyond laboratory conditions, and what were the main findings?
  8. What strategies do you envision to optimize or modify the SAPs to enhance their selectivity, functionality (e.g., nutrient release, antimicrobial activity)?

Author Response

Reviewer 2   

The manuscript describes a study on the preparation and characterization of superabsorbent polymers (SAPs) synthesized from cotton straw, acrylic acid, and acrylamide using 60Co gamma irradiation. The research focuses on sustainable SAP production by utilizing agricultural waste and evaluates potential applications in agriculture and medicine. The experimental approach includes both single-factor and orthogonal experiments to optimize SAP properties, and various analytical techniques (FTIR, XRD, TG, SEM) are used to support the findings. The study highlights the use of agricultural waste, a solvent-free radiation grafting method, and thorough material characterization. Results indicate high water absorbency and potential for practical application.

The manuscript would benefit from more detail on experimental reproducibility, statistical analysis, and a clearer comparison with other SAPs, especially in terms of biodegradability, cost, and environmental impact throughout synthesis and product use: Questions:

Point 1. Ensure that the quality of Figure 2 is clearly labeled i.e FTIR identifying the key functional groups, a more detailed discussion of the peak shifts and changes in intensity upon superabsorbent polymers (SAPs) formation would be beneficial. Figure 3. (a) XRD indexing and showing crystalline phases and Include the International Center for Diffraction Data (JCPDS) in the crystalline phases. Refer to(doi.org/10.1002/jctb.6657)and is recommended for citation.

Answer: Thank you. The important peaks of infrared have been labeled, and XRD analyzed them based on relevant literature.

Point 2. 2.  Figure3.(b) TG and DTG could be improved by better labeling.

Answer: Thank you for your suggestion, the pictures have been accurately labeled in the revised version.

Point 3. Could you provide detailed information on the reproducibility of SAP synthesis process, including the number of independent replicates performed and any statistical analyses undertaken?

Answer: Thank you for your suggestion, the scheme 1 have been accurately added for  “the reproducibility of SAP synthesis process,”, and  “each sample was tested three times[18, 19], standard error terms for Q from covariance (ANCOVA) using SPSS v. 19.0 (IBM Inc., Armonk, NY, USA).”  were added in the revised version.

Point 4. How does the long-term biodegradability and environmental safety profile of your SAPs compare to conventional synthetic alternatives, particularly regarding residual monomers and irradiation processes?

Answer: Thank you for your suggestion, our research team intends to conduct the next phase of research on how this highly absorbent resin naturally degrades in soil. Currently, this paper focuses on synthesis, performance and structural studies.

Point 5. Can you elaborate on the scalability and economic feasibility of utilizing cotton straw as a raw material, considering supply chain logistics and industrial-scale production challenges?

Answer: At present, the majority of cotton stalks are treated as agricultural waste. If this research can be industrialized, it will significantly reduce the consumption of petroleum.

Point 6. How do the mechanical properties (such as gel strength and elasticity) of your SAPs compare to those of commercially available products, especially under varying environmental conditions?

Answer:

Thank you for your suggestion, the next step will be to conduct field trials as a water-retaining agent. Once the field experiments are successful, industrial production will commence.

Point 7. Have you assessed the SAPs' practical performance, stability in real-world agricultural or medical settings beyond laboratory conditions, and what were the main findings?

Answer: Thank you for your suggestion, the next step will be to conduct a stability test for the high-absorbent resin used in medical supplies.

Point 8. What strategies do you envision to optimize or modify the SAPs to enhance their selectivity, functionality (e.g., nutrient release, antimicrobial activity)?

Answer: Thank you for your suggestion, it is planned to incorporate antibacterial substances such as zinc oxide, or to add fertilizers during the synthesis process to test their controlled-release properties.

Reviewer 3 Report

Comments and Suggestions for Authors

I think that this manuscript should be revised. Abstract, Introduction and Conclusion sections need to be revised, and more emphasis on scientific novelty of this research needs to be elaborated.  My general comments are presented in the attached file. 

Comments on the Quality of English Language

English language must be improved and sentencies revised. 

Author Response

Reviewer 3 

Title: Study on preparation and properties of Gels of homogenous Cotton Straw- acrylic acid - acrylamide by irradiation of 60Co gamma Using cotton straw waste to prepare adsorbent polymers is an eco-friendly, sustainable, low cost, renewable, and efficient solution for environmental cleanup. This biodegradable material helps reduce ecological pollution and dependence on non-renewable, synthetic polymers. It also adds value to agricultural waste that would otherwise be burned or discarded, contributing to cleaner air and less waste. The cellulose-rich structure of cotton straw can be modified to produce adsorbent polymers that can be used in agriculture, pharmaceuticals, and for the effective removal of pollutants like heavy metals and dyes from water. My general comments: Abstract section should be revised.

Point 1. Lines 13-17: The Sentence is too long, describing material used for modification, but also experimental technique in a single sentence. It has to be revised.

Lines 17-22: describe what should be in the Materials and Methods section, and only one part of the methods used.

I suggest that the Abstract section be revised to describe the modifications of cotton straw as waste materials, and that novel absorbent polymers and resins were prepared using NaOH/urea and crosslinking methods. A brief discussion about analytical techniques can be included in one sentence, followed by a discussion of the novelty of this research.

 Answer:  Thank you for your suggestion, In the revised version, the abstract has been rewritten.

Point 2. There should be a reference after Line 35.

Answer:  We are sorry that this is a mistake, The corresponding references has already been inserted there. in line 91-98 on page 3

Point 3. Line 36:  is used word „ resin“, but it is not clear if resin is an absorbent polymer produced. Is it resin or adsorbent resin, or adsorbent polymer from Line 12. 

Answer: We are sorry that this is a mistake, “Super absorbent gels” have replaced the “resin” in the revised version.

Point 4. Lines 36, 39: Sentences begin with „ These resins“, please revise those sentences and explain what are these resins

Answer: We are sorry that this is a mistake, In the revised version, I revised those sentences and explain what are these resins

Point 5. Line 40: the sentence has „ it has“, what is that“it“?

Line 42: ...which is rich...

Line 45: ....Several categories of what, please define briefly. Line 52 reference should be at the end.

Answer: We are sorry that this were mistakes, In the revised version, I revised those sentences as red words.

Point 6:  Line 60, 62 and 67: Please add a reference at the end of the sentence.

Answer:  Thank you for your suggestion, the corresponding references has already been inserted there in the revised version.

Point 7: Lines 71-77: discuss the same ideas as lines 56-60. This paragraph should be removed after line 60.

Answer: Thank you for your suggestion, the corresponding paragraph has already been removed there in the revised version.

Point 8: Line 88: Please describe what CS byproducts are.

Line 94: Please add a reference at the end of the sentence that discusses the utilization of waste cotton stalks.

Answer: We are sorry that this were mistakes, Line 88:  CS was replaced by “Cotton straw” in the revised version. Line 94: we have add the reference at the end of the sentence in the revised version.

Point 9: Lines 109-113: should be moved prior to line 103. 

Lines 120-126: should be moved prior to line 105.

Answer: We are sorry that this is a mistake, it were modified in revised version.

Point 10: Lines 130 and 133: Please discuss the novelty of this research and why it differs from similar experiments done in this field.

Answer: Thank you for your suggestion, “In this study, cotton stalks were dissolved in different concentrations of alkaline solution to obtain homogeneous stalk solutions without crystalline regions. This process enabled the large molecular fibers in the stalks to undergo graft copolymerization with monomer molecules. The copolymerization reaction was initiated by cobalt radiation, which saved chemical initiators and reduced pollution.” were added in the revised version.

Point 11: Line 136-142: This description of NaOH/urea aconcnetration for obtaining different CS samples should be put in Table .

Answer: Thank you for your suggestion, this description of NaOH/urea aconcnetration for obtaining different CS samples should be put in Table 1 . were added in the revised version.

Point 12:  Authors should discuss whether this is used to dissolve cellulose, as decrystallization of cellulose is not a sufficient term. Delamination, swelling of cellulose chains, and dissolution are stages in which cellulose fiber transfers to dissolved polymer. Please revise this section.

Answer: Thank you for your suggestion, the corresponding paragraph has already been added there in the revised version.

Point 13: Line 182: Please revise the term“ reaction product space. “?

Answer: We are sorry that this is a mistake, it were modified in revised version.

Point 14: What is monomer in Line 178 and Figure 1. Please describe this in the text.

Answer: We are sorry that this is a mistake, it were modified in revised version.

Point 15: Line 191: It is difficult to follow this 191 -193 without previously reading the Materials and methods section.

Answer: We are sorry that this were mistakes, it were modified in revised version.

Methods and Materials should be placed in the fourth section, and Results and Analysis in the second section, as per the editorial department's format requirements. I apologize for any inconvenience this may cause.

Point16: Lines 208-224: Discussion of Figure 1 should be above Figure 1.

Lines 2016-223: It is not clear whether the authors still refer to Fig.1.

Answer: We are sorry that this is a mistake, that were modified in revised version.

Point17: Line 227: Word „does“ should be replaced with the word „dose“.

Line 304: Please define which sample is presented in Fig.4, as there is only one SEM image.

Answer: Line 227: does“ has been replaced with the word „dose“. in the revised version.

Line 304: We are sorry that this is a mistake, it were modified in revised version.

Point19: The conclusion section should be revised and instead of being a short version of Methods section, as presently is, should discuss the overall results of this research in light of novel scientific achievements.

Point20: The Materials and Methods section should be placed prior to the Experimental section. In the Materials section, it should be stated where the chemicals were purchased from.

Answer: Thank you for your suggestion, this is a paper written according to the fixed template of the journal. The chemical reagent purchasing company has marked the red font in the Materials and Methods section.

Point20: I suggest that authors draw a schematic presentation of the methods used in this research.

Answer: Thank you for your suggestion, the corresponding scheme 2 has already been added there in the revised version.

Point 21: Line 343: Please make a schematic drawing or refer to the device where cooking was performed.

Answer: Line 343: Thank you for your suggestion, the corresponding scheme 1 has already been added there in the revised version.

Point 22: Line 363: Please replace the word „ certain“ with an adequate amount used or refer to the table that is now missing in the manuscript and needs to be made.

Answer: Line 363: Thank you for your suggestion, in this study, the reagents and raw materials used in single-factor experiments are variables, which require orthogonal experiment optimization to improve conditions. Therefore, specific ratio data cannot be provided at this point.

Point23: Lines 372-392 : How were prepared samples for SEM?

Answer: We are sorry that this is a mistake, it were modified as 4.5(b) in revised version.

Reviewer 4 Report

Comments and Suggestions for Authors

Authors elaborate a process of graft copolymerization induced by 60Co radiation to convert cotton straw into a superabsorbent polymer material, for which numerous technical applications exist. The context of cotton straw recycling and the scientific background are well developed in introduction. I have no reservation on the design and the conduction of the experimental study. On the other hand, acceptation of the manuscript will only be possible after improvements and completions of the text, under consideration of following points.

  1. There is a lack of figures in this manuscript, which compromises the clarity and the visibility of the work. Author should include a schematic representation of the successive steps of the elaboration of superabsorbent polymer from cotton straw. Additionally, the developed formula of the reagents of the process, in particular AA, Am and MBA, should be shown.

  1. Lines 136-153. Authors try seven pretreatments with the aim to select the best one. But how were these seven pretreatment conditions chosen? Please, insert few sentences explaining the idea behind the NaOH/urea ratio variation and what was expected with the tests CS5 to CS7.

2a Line 149. What means "PCS" in "cellulose PCS"? Please define.

  1. Figure 1. The graphs are pixelized and too small. The subscripts in graph titles and curve labels are not readable.

3a. Legend of figure 1. I guess the graph curve labels are "Qd" and "Qs". These quantities must be defined in the legend of figure 1.

  1. Line 210. It should be: "the range of 50% to 80%", not "…100%"

  1. Line 285-286. This is an abuse of language. The narrow XRD peak at 20-23° is a reflection demonstrating the presence of crystallites, in contrast to the purely amorphous graft copolymer. XRD gives no direct information on hydrogens bonds but you may assume that crystallites are characterized by stronger hydrogen bonds.

  1. Lines 20-21 "AA" and "Am" are used without definition in abstract

  1. Lines 314 to 321. The conclusions mainly summarize the results of optimization. They should be strengthened and address the scientific questioning in introduction. One question naturally arising is on the optimal 7:1 monomer to CS ratio: could a material with such a low CS content really represent an outlet for this agricultural by-product? Please, introduce some comments on this point, as well as on the potential interests for applications of the studied process.
Comments on the Quality of English Language

The Quality of English Language is overall good but careful rereading is necessary.

- At few places there are problems of syntax obscuring the meaning of the sentence, for instance in lines 212-213 "resulting in… is affected".

- There are also misprints. Note that "dose" is systematically misspelled in "does".

Author Response

Reviewer 4

Authors elaborate a process of graft copolymerization induced by 60Co radiation to convert cotton straw into a superabsorbent polymer material, for which numerous technical applications exist. The context of cotton straw recycling and the scientific background are well developed in introduction. I have no reservation on the design and the conduction of the experimental study. On the other hand, acceptation of the manuscript will only be possible after improvements and completions of the text, under consideration of following points.

 Point1: There is a lack of figures in this manuscript, which compromises the clarity and the visibility of the work. Author should include a schematic representation of the successive steps of the elaboration of superabsorbent polymer from cotton straw. Additionally, the developed formula of the reagents of the process, in particular AA, Am and MBA, should be shown.

Answer: Thank you for your suggestion, the corresponding scheme 2 has already been added there in the revised version.

Point 2: Lines 136-153. Authors try seven pretreatments with the aim to select the best one. But how were these seven pretreatment conditions chosen?

Answer:  Choose the appropriate alkali concentration and straw dissolution method based on the literature.

Point 2: Please, insert few sentences explaining the idea behind the NaOH/urea ratio variation and what was expected with the tests CS5 to CS7.

Answer: Thank you for your suggestion, at 2.1, the corresponding “The alkaline solution contains highly polar hydroxide ions and sodium ions. It can break the hydrogen bonds between cellulose molecules, causing cellulose to lose its crystalline structure, dislodge the supermolecular coiling structure of cellulose, and separate the long chains of molecules. This results in the swelling and dissolution of cellulose which fiber transfers to dissolved polymer, improve the efficiency of graft copolymerization[37].” has already been added there in the revised version.

Point2: 2a Line 149. What means "PCS" in "cellulose PCS"? Please define.

Answer: We are sorry that this is a mistake, it were modified in revised version.

Point3: Figure 1. The graphs are pixelized and too small. The subscripts in graph titles and curve labels are not readable.

Answer: We are sorry that this is a mistake, it were modified in revised version.

3a. Legend of figure 1. I guess the graph curve labels are "Qd" and "Qs". These quantities must be defined in the legend of figure 1.

Answer: We are sorry that this is a mistake, it were modified in revised version.

Point4: Line 210. It should be: "the range of 50% to 80%", not "…100%"

Answer: We are sorry that this is a mistake, at 2.2.3, it were modified in revised version.

Point 5: Line 285-286. This is an abuse of language. The narrow XRD peak at 20-23° is a reflection demonstrating the presence of crystallites, in contrast to the purely amorphous graft copolymer. XRD gives no direct information on hydrogens bonds but you may assume that crystallites are characterized by stronger hydrogen bonds.

Answer: Thank you for your suggestion, “Cellulose is a natural high molecular weight polysaccharide compound. The weak intermolecular hydrogen bonding interactions and polar charge distribution lead to the formation of a supramolecular form[39]. After being treated with the corresponding chemical or physical methods, the supramolecular entanglement state within the molecules changes, the hydration capacity is enhanced, and after graft co-polymerization with small chemical molecules, the supramolecular structure is further stretched, making the amorphous regions richer[40]”. were added in revised version.

Point 6: Lines 20-21 "AA" and "Am" are used without definition in abstract

Answer: We are sorry that this is a mistake, for abstract, that were modified in revised version.

Point7: Lines 314 to 321. The conclusions mainly summarize the results of optimization. They should be strengthened and address the scientific questioning in introduction. One question naturally arising is on the optimal 7:1 monomer to CS ratio: could a material with such a low CS content really represent an outlet for this agricultural by-product? Please, introduce some comments on this point, as well as on the potential interests for applications of the studied process.

Answer:  Thank you for your suggestion, “Straw cellulose is a natural macromolecular polysaccharide, which is composed of glycosidic bonds linking sugar units into chain-like polymers. As a resin matrix, it is easily degradable because the formation of a three-dimensional network structure requires the cross-linking of small molecule monomers. With less cellulose usage, it will be used as a fertilizer controlled-release agent and water retention agent for field experiments in the later stage. The rate of salt water absorption in this experiment reflects to a certain extent that it can maintain a certain amount of ionic water in the field environment.”  were added in the in revised version for the conclusion.

Point8:  Comments on the Quality of English Language

The Quality of English Language is overall good but careful rereading is necessary.

At few places there are problems of syntax obscuring the meaning of the sentence, for instance in lines 212-213 "resulting in… is affected".

Answer: This paper will utilize MDPI's English language polishing service to improve grammatical issues.

Point 9: - There are also misprints. Note that "dose" is systematically misspelled in "does".

Answer: We are sorry that this is a mistake, that were modified in revised version.

Round 2

Reviewer 1 Report

Comments and Suggestions for Authors

Accept

Reviewer 2 Report

Comments and Suggestions for Authors

The authors addressed my suggestions.

Reviewer 3 Report

Comments and Suggestions for Authors

The authors have answered my comments and improved the quality of the manuscript. 

Reviewer 4 Report

Comments and Suggestions for Authors

Authors adequately revised/completed their manuscript, which can now be accepted in present form.